# Epidemiological Factors Associated with Gross Diagnosis of Pulmonary Pathology in Feedyard Mortalities

**DOI:** 10.3390/vetsci10080522

**Published:** 2023-08-14

**Authors:** Eduarda M. Bortoluzzi, Brad J. White, Paige H. Schmidt, Maddie R. Mancke, Rachel E. Brown, Makenna Jensen, Phillip A. Lancaster, Robert L. Larson

**Affiliations:** 1Department of Anatomy and Physiology, Kansas State University, Manhattan, KS 66506, USA; bortoluzzi@vet.k-state.edu; 2Beef Cattle Institute, Kansas State University, Manhattan, KS 66506, USA; phschmidt@vet.k-state.edu (P.H.S.); mmancke@vet.k-state.edu (M.R.M.); rebrown@vet.k-state.edu (R.E.B.); makenn1@vet.k-state.edu (M.J.); palancaster@vet.k-state.edu (P.A.L.); rlarson@vet.k-state.edu (R.L.L.)

**Keywords:** acute interstitial pneumonia, bronchopneumonia with an interstitial pneumonia, bronchopneumonia, epidemiology, feedyard cattle, bovine respiratory disease (BRD)

## Abstract

**Simple Summary:**

Pulmonary pathology is a common finding in feedyard cattle, with bronchopneumonia and interstitial pneumonia being two of the most frequent diagnoses. Often cattle have either bronchopneumonia or interstitial pneumonia, but recently they have been observed to display pathology associated with both pathological processes. This study aimed to describe the agreement between necropsy and feedyard clinical diagnoses and describe the epidemiologic characteristics of feedyard pulmonary pathologies.

**Abstract:**

Respiratory disease continues to be the major cause of mortality in feedyard cattle, with bronchopneumonia (BP) and acute interstitial pneumonia (AIP) as the two most common syndromes. Recent studies described a combination of these pathological lesions with the presence of AIP in the caudodorsal lungs and BP in the cranioventral lungs of necropsied cattle. This pulmonary pathology has been described as bronchopneumonia with an interstitial pneumonia (BIP). The epidemiological characteristics of BIP in U.S. feedyard cattle are yet to be described. This study’s objectives were to describe the agreement between feedyard clinical and necropsy gross diagnosis and to characterize epidemiological factors associated with four gross pulmonary diagnoses (AIP, BIP, BP, and Normal pulmonary tissue) observed during feedyard cattle necropsies. Systemic necropsies were performed at six feedyards in U.S. high plains region, and gross pulmonary diagnoses were established. Historical data were added to the dataset, including sex, days on feed at death (DOFDEATH), arrival weight, treatment count, and feedyard diagnosis. Generalized linear models were used to evaluate epidemiological factors associated with the probability of each pulmonary pathology. Comparing feedyard clinical diagnosis with gross pathological diagnosis revealed relatively low agreement and the frequency of agreement varied by diagnosis. The likelihood of AIP at necropsy was higher for heifers than steers and in the 100–150 DOFDEATH category compared with the 0–50 DOFDEATH (*p* = 0.05). The likelihood of BIP increased after the first treatment, whereas the DOFDEATH 0–50 category had a lower likelihood compared with the 150–200 category (*p* = 0.05). These findings highlight the importance of necropsy for final diagnosis and can aid the development of future diagnosis and therapeutic protocols for pulmonary diseases.

## 1. Introduction

Respiratory diseases are the main cause of morbidity and mortality in feedyard beef cattle [1,2,3]. Bronchopneumonia (BP) is the most common syndrome and results from multiple pathogens and risk factors associated with the bovine respiratory disease complex [4,5,6,7]. Acute interstitial pneumonia (AIP) is less frequent than BP, but AIP is very impactful due to the relatively high case fatality risk. The clinical distinction between BP and AIP is made based on clinical signs and epidemiological characteristics of affected cattle. Distinguishing these syndromes influences therapeutic decisions as BP often contains a bacterial pathogen warranting antimicrobial therapy, whereas AIP may not have a bacterial component [8,9]. Epidemiologic factors differ between syndromes, and BP is reported to occur early in the feeding phase and, more commonly, in steers [10,11]. In contrast, AIP is reported to affect cattle longer than 60 days on feed and, more commonly, in heifers [12,13,14]. Another important difference is the sudden onset of respiratory symptoms in AIP compared with BP cases [15]. While these distinctions are important, cattle afflicted with BP and AIP will show similar clinical signs, and little is known about the agreement between the feedyard clinical and gross necropsy diagnoses.

Recently, researchers reported the simultaneous presence of AIP and BP in deceased feedyard cattle. A new classification has been suggested as bronchopneumonia with an interstitial pneumonia (BIP) due to the presence of lesions compatible with AIP in caudodorsal lobes and BP on cranioventral lobes [16,17,18,19]. In previous studies when lesions consistent with BIP were found during necropsy, the final diagnosis or cause of death was limited to AIP or BP only, and it was dependent on the pathology that affected the majority of the lung [14,20,21]. The newly proposed diagnosis of BIP appears to be important because BIP was reported in one-third of necropsied feedyard cattle presenting lung lesions [17]. Epidemiological characteristics of BIP cases were reported in Canadian feedyard cattle [16]. However, little is known about the epidemiological characteristics of this ascending pathology in U.S. feedyard cattle compared with standalone diagnoses of AIP and BP. An improved understanding of epidemiological comparisons among pulmonary pathologic syndromes could lead to improved and targeted disease intervention techniques.

Previous research has evaluated risk factors for the onset of pulmonary disease in both cohorts and individuals; however, little research has compared risk factors for the development of specific pulmonary syndromes [22,23,24,25,26]. Therefore, the aims of this study were to describe the agreement between feedyard treatment diagnosis and necropsy gross pulmonary diagnosis and characterize the epidemiology factors associated with four gross pulmonary diagnoses (AIP, BIP, BP, and normal pulmonary tissue) observed during feedyard cattle necropsies.

## 2. Materials and Methods

This research in deceased animals was deemed exempt by the Kansas State University IACUC (Institutional Animal Care and Use Committee).

### 2.1. Case Collection and Gross Diagnosis

This study was designed as a cross-sectional observational study. Feedyard cattle fatalities were systemically necropsied between June and July 2022 at six feedyards in the U.S. high plains region. Necropsies were performed by four trained necropsy technicians, and gross pulmonary diagnoses were determined conjointly with a veterinarian. Cattle enrolled in this study were those presenting one of the following post-mortem gross pulmonary diagnoses: acute interstitial pneumonia (AIP), bronchopneumonia (BP), bronchopneumonia with an interstitial pneumonia (BIP), and normal pulmonary tissue (Normal).

Lungs grossly diagnosed with AIP exhibited pulmonary lobules with colors varying from light pink to dark red, diffuse overinflated lobes with interlobular edema and emphysema, consistent with a “checkerboard” pattern [20]. Lungs grossly diagnosed with BP presented with a variety of lesions, including lung consolidation, pulmonary abscesses, firm or rubbery texture, and fibrinous interlobular material [20]. The BIP grossly diagnosed lungs presented with features compatible with BP in the cranioventral lung lobes and features compatible with AIP in the caudodorsal lung lobes [16,19]. Normal lung tissue was presented as deflated lungs with pale pink coloration and absence of any gross pulmonary lesions. Cattle may have presented with additional gross pathological lesions in other organ systems, but cases were classified based on presence or absence of pulmonary pathologies as described above.

### 2.2. Retrospective Health Data

Retrospective individual health data were sourced from the feedyards where cattle were housed prior to mortality and necropsy. Data were matched using individual animal identification and lot numbers. Variables collected included arrival date, cohort arrival weight, number of individual-animal treatment events, date of treatment events, feedyard clinical diagnoses at the time of treatment, treatments administered, mortality date, feedyard identifier, and sex. Feedyard diagnoses were recorded by trained feedyard personnel based on antemortem clinical signs. Clinical signs consistent with bovine respiratory disease (BRD) included depression, anorexia, and increased respiration rates. Diagnosis of AIP was based on similar clinical signs with rapid onset and severe respiratory distress, often resulting in increased expiratory effort. Diagnosis of illness was initially performed on individuals within the cohort and confirmed and recorded following chute side evaluation by trained personnel. Clinical diagnoses were grouped into 5 categories: AIP: the animal was clinically diagnosed one or multiple times for AIP only; BP: the animal was clinically diagnosed one or multiple times for BP only; AIP + BP: the animal was clinically diagnosed at least once for AIP and at least once for BP, regardless of order; AIP or BP + OTHER: the animal was clinically diagnosed for AIP or BP at least once and clinically diagnosed with another pathology at least once; Not Diagnosed: when the animal was not diagnosed while alive; and OTHER. The OTHER category comprised a variety of diagnoses, including abortion, abscess, bleeder, bloat, buller steer syndrome, calver, lameness, diphtheria, downer, ear infection, footrot, and injury.

### 2.3. Statistical Analysis

Data manipulation was performed in Excel (Microsoft Excel, Microsoft 365^®^) and RStudio (R Core Team, 2022) software. For descriptive purposes, tables comparing feedyard treatment diagnoses and necropsy gross diagnoses were made using Excel. The ‘Kappa2’ function of the ‘irr’ package of RStudio was used to calculate Cohen’s Kappa index agreements between AIP, BIP, and BP categories of necropsy and feedyard diagnosis. Three data sets were created for these data for this analysis, one for each category (AIP, BIP, and BP), where the diagnosis was classified binomially as present (1) and absent (0). Category BIP was created for feedyard diagnosis when cattle were treated at least one time for AIP and one time for BP, regardless of order.

Four new variables were created to binomially categorize each necropsy gross diagnosis of AIP, BP, BIP, and Normal as 1 (present) and 0 (absent). New variables were created to categorize the treatment count per animal (TXCNT; 1, 2, and >3), antibiotic treatment count per animal (ABTXCNT; 1, 2, and >3), and anti-inflammatory treatment count per animal (AITXCNT; 1, 2, and >3). The total days on feed until first treatment (DOF1TX) was calculated by subtracting first treatment data by arrival date, and then it was categorized into 4 categories (0–50, 50–100, 100–150, and 150–200 days). Days on feed at death (DOFDEATH) was calculated by subtracting the death date from the arrival date and was categorized into 4 categories (0–50, 50–100, 100–150, and 150–200 days). Arrival weight (INWT) was categorized into 4 categories (<272, 272–317, 317–363, and >363 kg). The variables DOF1TX, DOFDEATH, and INWT were categorized to avoid violating the assumption of linearity.

Four generalized linear models were used to analyze the data using the ‘glm’ function of the ‘stats’ package of RStudio. Correlation coefficients were used to find collinearity amongst variables. Variables ‘AITXCNT’ and ‘ABTXCNT’ were found to be highly correlated with ‘TXCNT’ and therefore were not included in the model. The first model accounted for the probability of an animal being diagnosed with AIP at necropsy and included fixed effects of sex, ‘INWT’, ‘TXCNT’, ‘DOFDEATH’, and ‘feedyard’. The subsequent three models had the same fixed effects, however, accounting for the probability of each one of the other diagnoses (BP, BIP, and Normal). The significance level was set at *p* < 0.05.

## 3. Results

### 3.1. Population Descriptive Statistics

Between June and July 2022, 402 cattle in the U.S. high plains region were necropsied by study technicians, and of those, 360 were enrolled in this study based on pulmonary pathology classified at gross necropsy as AIP, BP, BIP, or Normal. Forty-two cases were excluded because they had some other lung lesion that was not compatible with AIP, BIP, BP, or Normal. A total of 3 of the 360 cases had incomplete historical data and were removed from the analysis. Heifers accounted for 70.6% (252), and steers accounted for 29.4% (105) of the cases. These proportions are consistent with cattle demographics within the feedyard enrolled in the study. The median body weight at arrival was 347 kg (IQR = 312–368). At necropsy, 11.2% (40) presented gross lesions compatible with AIP, 40.6% (145) with BP, 39.8% (142) with BIP, and 8.4% (30) with normal pulmonary tissue. A total of 27.2% (97) of the animals never had previous treatment, 29.4% (105) were treated once, 18.8% (67) were treated twice, and 24.6% (88) were treated three or more times. The median DOF1TX was 52 (IQR = 21–111), and the median DOFDEATH was 92 days (IQR = 47–132). Antibiotics were used as a treatment at least once in 70.3% of the animals, and steroid anti-inflammatories were used at least once in 50.1% of the animals. Most of the AIP cases diagnosed during necropsy were heifers (85%; Table 1). The same trend was observed for BIP (69%) and BP (71%).

### 3.2. Necropsy Gross Diagnoses and Feedyard Diagnosis

During clinical identification for treatments, the feedyards assigned a diagnosis for each animal. Some animals were treated multiple times and received different diagnoses, such as cattle treated at least one time for AIP and one time for BP (AIP + BP). An animal could also have been treated at least one time for AIP or BP and at least one time for OTHER diagnoses (BP or AIP + OTHER). At treatment, 13.2% (47) of the animals were diagnosed with AIP, 34.5% (123) with BP, 5% (18) with AIP + BIP, 10.6% (38) with BP or AIP + OTHER, 9.5% (34) with OTHER, and 27.2% (97) were not diagnosed or treated while alive. When comparing the diagnoses found at necropsy with the feedyard diagnoses, AIP had an agreement of 22.5% and a Cohen’s kappa of 0.098 (Figure 1). When we compared the AIP + BP group with the necropsy diagnosis of BIP, there was only a 6.3% (9/142) agreement and a Cohen’s kappa of 0.025. Most cases that had BIP as a diagnosis during necropsy had been assigned a BP feedyard diagnosis (58/142). The agreement between necropsy and feedyard diagnoses for BP was 27.6% (40/145) and had a Cohen’s kappa of −0.118.

Raw data of temporal distributions of DOF1TX based on necropsy pulmonary gross diagnoses and feedyard diagnoses are depicted in Figure 2A,B, respectively. The AIP diagnosis made during necropsy had a median of 81 DOF1TX and a median of 113 when considering the feedyard diagnoses. Similar trends are observed for a BIP diagnosis at necropsy and AIP + BP feedyard diagnoses.

### 3.3. Probabilities Models

#### 3.3.1. Acute Interstitial Pneumonia

A generalized linear model was fit to identify potential relationships between covariates of interest (sex, INWT, TXCNT, DOFDEATH, and feedyard) and the binomial response of AIP. Sex and DOFDEATH were significantly (*p* < 0.05) associated with the probability of AIP at necropsy. The model also accounted for the effects of INWT, TXCNT, and feedyard, but they were not significant. Heifers showed a higher probability of being diagnosed as AIP at necropsy compared with steers (*p* < 0.05; Figure 3A). The probability of an AIP diagnosis was higher in the DOFDEATH of the 100–150 category compared with 0–50; however, no other categories significantly differed (*p* > 0.05; Figure 3B).

#### 3.3.2. Bronchopneumonia with an Interstitial Pneumonia

A generalized linear model was fit to account for potential associations between covariates (sex, INWT, TXCNT, DOFDEATH, and feedyard) and the binomial response of BIP. Treatment count was associated (*p* = 0.05) with the probability of BIP, whereas DOFDEATH tended (*p* = 0.06) to be associated with the probability of a BIP diagnosis at necropsy. The model also accounted for the effects of INWT, which was not significant, and feedyard. No differences were found in the probabilities of diagnosing BIP between heifers and steers (*p* > 0.05). However, the probability of BIP was greater in cattle treated 1 or >3 times than in those never treated (*p* < 0.05; Figure 4A). Cattle with DOFDEATH in the 0–50 category had a lower probability compared with cattle with DOFDEATH of 150–200; however, no other differences were identified between categories (Figure 4B).

#### 3.3.3. Bronchopneumonia

A generalized linear model was fit to account for the binomial response of BP at necropsy and the fixed effects of sex, INWT, TXCNT, DOFDEATH, and feedyard. The probabilities of BP diagnoses at necropsy were not associated with any fixed effects (*p* > 0.05).

#### 3.3.4. Normal Pulmonary Tissue

A generalized linear model with the same covariates as the other models was attempted to determine associations with the probability of Normal pulmonary tissue. This model did not converge due to the small sample size and, therefore, will not be discussed.

## 4. Discussion

Pulmonary disease is an important contributor to feedyard mortality, and the appropriate classification of disease types can improve strategies to mitigate deleterious disease events. This study evaluated necropsy findings at six U.S. high-plains feedyards and identified three primary classifications of pulmonary lesions, with BP (40.6%) and BIP (39.8%) representing most cases. The BIP gross pulmonary diagnosis consists of lesions associated with both BP and AIP, and while anecdotal reports and some scientific manuscripts have described this lesion pattern, relatively few have acknowledged this as a separate syndrome. A small proportion of the population had normal pulmonary tissue (8.4%), and AIP was identified in the remainder of the population (11.2%). The proportion of feedyard cattle presenting with each pulmonary lesion category in this study follows the trend of previously studied cohorts [16]. Most cattle (72.8%) had at least one antemortem treatment, and agreement among clinical and gross necropsy diagnoses was relatively low and varied by gross necropsy diagnosis. The temporal pattern of initial treatment illustrated a distinct pattern based on clinical diagnosis, with AIP diagnosed later in the feeding period; however, when evaluating gross pathological diagnosis, AIP cases were treated throughout the feeding period. Comparing the probabilities of each gross diagnosis revealed that risk factors differed based on a diagnosis of BP, BIP, or AIP.

Relatively little data exists comparing clinical and gross pathological diagnoses in feedyard mortalities. Our results showed interesting disagreements: most (72%) AIP cases identified at necropsy were clinically diagnosed as something other than AIP. In addition, 22 AIP feedyard diagnoses presented lesions consistent with BIP during necropsy, suggesting that BIP is likely misdiagnosed as AIP. This misdiagnosis could be related to BIP occurring later in the feeding period or the rapid onset of disease symptoms similar to those reported for AIP. This possible misdiagnosis agrees with the description of AIP and past/chronic pulmonary injuries’ simultaneous occurrence described by other researchers [15,27,28]. Most cattle clinically diagnosed with BP (108/123) were grossly diagnosed with BP or BIP, indicating a BP component to the pulmonary pathology. This is potentially an important distinction as BP may have several etiologic agents present, and the mainstay of therapy is antimicrobial therapy. Distinguishing BP from AIP can be clinically important, but based on these results discriminating between the two syndromes, antemortem is challenging.

The temporal distribution of clinical treatment during the feeding period provides more insight into disease timing. Evaluating AIP cases shows a discrepancy of days on feed at first treatment between clinical and necropsy diagnoses. Cattle clinically diagnosed with AIP were clustered between 80–140 DOF1TX (median 113), which is similar to previous reports of this syndrome occurring late in the feeding phase [27,29,30]. In contrast, diagnosis of AIP based on gross pulmonary pathology illustrated a more even distribution through the feeding period with a median of 81 DOF1TX. This discrepancy could indicate that in addition to clinical signs, feedyard diagnosticians are utilizing demographic factors (DOF) to determine diagnosis; however, based on pulmonary pathology, AIP cases are not discretely limited to only later in the feeding period. The temporal distribution of BP cases was similar based on either clinical or gross pulmonary diagnosis. The first treatment for BP occurred early in the feeding period corroborating the epidemiological characteristics of BP previously described [3,11,22]. Discrepancies among temporal distributions based on gross vs. clinical diagnosis illustrate the value of gross pulmonary pathology to enhance overall diagnostic accuracy.

The Cohen’s kappa index agreements were poor in this study; however, it is important to note this analysis did not account for diagnosis order or which was the last feedyard diagnosis. Differences between percent agreement and Cohen’s kappa likely exist because Cohen’s kappa considers the prevalence of the outcome in the dataset. For example, the prevalence of BIP in the feedyard diagnosis was very low, 18 out of 357, likely affecting the index agreement. Thus, the limitations of this dataset might have driven these poor index agreements.

The generalized linear models developed allowed the determination of characteristics associated with the likelihood of each pulmonary pathology accounting for important effects such as sex, arrival weight, feedyard, number of treatments, and days on feed at death. The epidemiologic characteristic of pulmonary AIP lesions at necropsy was consistent with prior studies. The likelihood of AIP occurrence was higher in heifers than steers and lower in cattle of 0–50 DOFDEATH compared with cattle of 100–150 DOFDEATH [15,27,29]. The probability of AIP pulmonary diagnosis did not linearly increase with DOFDEATH, and this could be due to relatively small numbers of mortalities. Seasonality has been reported as a risk factor for AIP cases, with higher risk reported for summer months; however, this could not be evaluated in this trial due to case enrollment only during the summer months. Further investigation would be useful to better understand the timing of AIP and to investigate the rationale for increased risk in heifers.

One relatively novel area for this study was the classification of BIP as the combination of gross pulmonary pathologies of BP and AIP. Previous research in the U.S. and Canada showed that BIP was present in about two-thirds of feedyard cattle presenting pulmonary lesions at necropsy [16,17]. Nevertheless, investigations of the epidemiological characteristics associated with BIP are scarce for U.S. feedyard cattle. Results from the generalized linear models did not show an association between the increased probability of BIP and male cattle, as previously reported by Haydock et al. [16]. This finding could be related to the lower number of steers in the current study’s population or to the fact that the current study included all potential demographic factors into a single generalized linear model. Previous work illustrated a higher number of average treatments for BIP cases compared with both AIP and BP; however, current results illustrated an increased risk of BIP in cattle treated at least once for any type of pathology. This finding could be important, as at least one previous diagnosis and treatment was associated with a higher probability of BIP compared with other syndromes. Similar to AIP, BIP risk was lower in DOFDEATH 0–50 compared with 150–200 DOFDEATH. The greater likelihood of BIP in cattle late in the feeding period with more than one treatment could indicate that cattle might experience one type of pneumonia first and then a second one leading to death. Woolums [27] described the presence of bacterial pathogens isolated from the pulmonary tissue of cattle with AIP, showing that a possible bacterial infection could be causing BP in cattle already impacted by interstitial pneumonia or that the bacterial presence promotes a significant inflammatory response causing interstitial pneumonia. These findings indicate several differences in risk factors associated with BIP compared with risk factors for BP and AIP and highlight the importance of evaluating this as a potentially separate syndrome. Research to further understand BIP development to evaluate if BP occurs first in cranioventral lungs and then is followed by AIP in caudodorsal lungs, or if AIP develops in the caudodorsal lungs and then is followed by BP in the cranioventral lungs, or if both lesions develop at the same time would be valuable. Improved knowledge of the pathological progression may lead to improved diagnostic and therapeutic interventions for BIP cases.

## 5. Conclusions

Results indicate three pulmonary pathologies (BP, BIP, and AIP) are common in feedlot mortalities, and each pathology differs in epidemiological characteristics and temporal distributions. Clinical diagnosis and gross pathological diagnosis illustrate discrepancies in ante- and post-mortem signs, highlighting the value of collecting and comparing all available diagnostic information. Previous research has illustrated that syndromes such as AIP and BIP occur late in the feeding phase; however, these results indicate a wide distribution of the timing of the onset of cases after arrival. Risk factors vary between BP, BIP, and AIP, illustrating that each syndrome may have differing pathogenesis. The BIP cases are rarely reported as a separate syndrome, but these results indicate there may be some value in discriminating between this type of pulmonary lesion as compared with BP and BIP. These findings are important to develop future diagnosis and treatment protocols for pulmonary syndromes in feedyard cattle.

## Figures and Tables

**Figure 1 vetsci-10-00522-f001:**
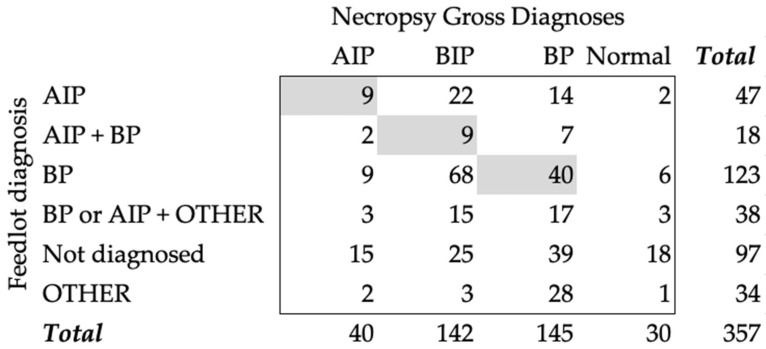
Count table of necropsy gross diagnoses and feedyard diagnoses given during treatment. Gray boxes represent the agreement counts between the two diagnoses. AIP—acute interstitial pneumonia; BIP—bronchopneumonia with an interstitial pneumonia; BP—bronchopneumonia; Normal—normal pulmonary tissue; Other—a variety of diagnoses, including abortion, abscess, bleeder, bloat, buller steer syndrome, calver, lameness, diphtheria, downer, ear infection, footrot, and injury; AIP + BIP—cattle diagnosed with AIP and BP at different times, regardless of order; BP or AIP + OTHER—cattle diagnosed with BP or AIP at least one time and diagnosed with another condition in separated times; Not Diagnosed—cattle never diagnosed in the feedyard (never treated).

**Figure 2 vetsci-10-00522-f002:**
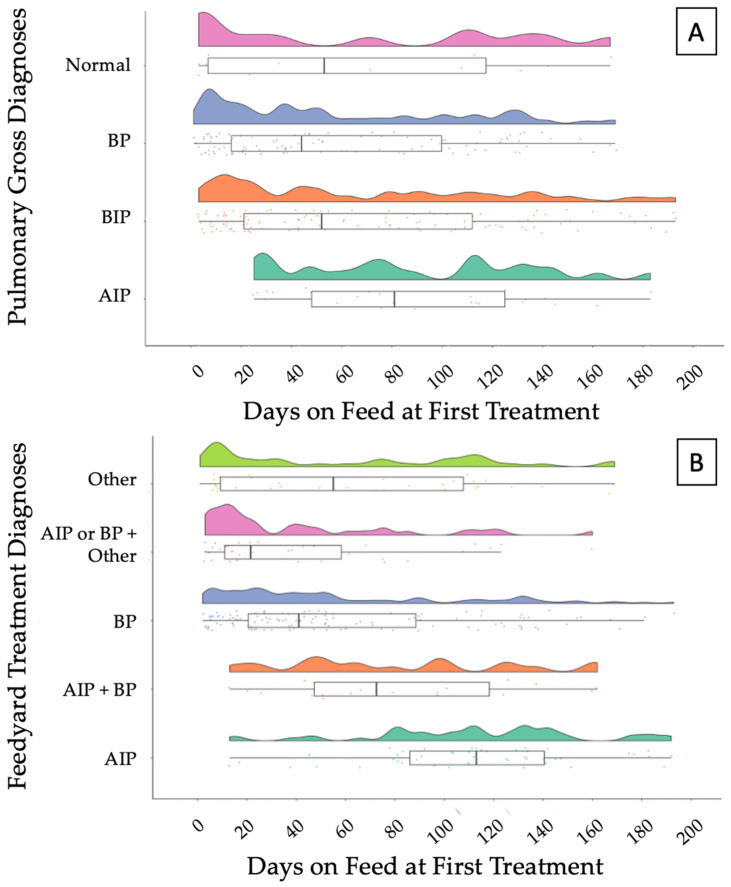
Raincloud plot depicting (**A**) the temporal distribution of necropsy pulmonary gross diagnoses and (**B**) feedyard treatment diagnoses over days on feed at first treatment (DOF1TX). Box and whisker represent the upper and lower quartiles, and the heavy line within the box represents the median. AIP—acute interstitial pneumonia; BIP—bronchopneumonia with an interstitial pneumonia; BP—bronchopneumonia; Normal—normal pulmonary tissue; OTHER—a variety of diagnoses, including abortion, abscess, bleeder, bloat, buller steer syndrome, calver, lameness, diphtheria, downer, ear infection, footrot, and injury. AIP + BIP—cattle diagnosed with AIP and BP at different times, regardless of order; BP/AIP + OTHER—cattle diagnosed with BP or AIP at least one time and diagnosed with other conditions at separate times.

**Figure 3 vetsci-10-00522-f003:**
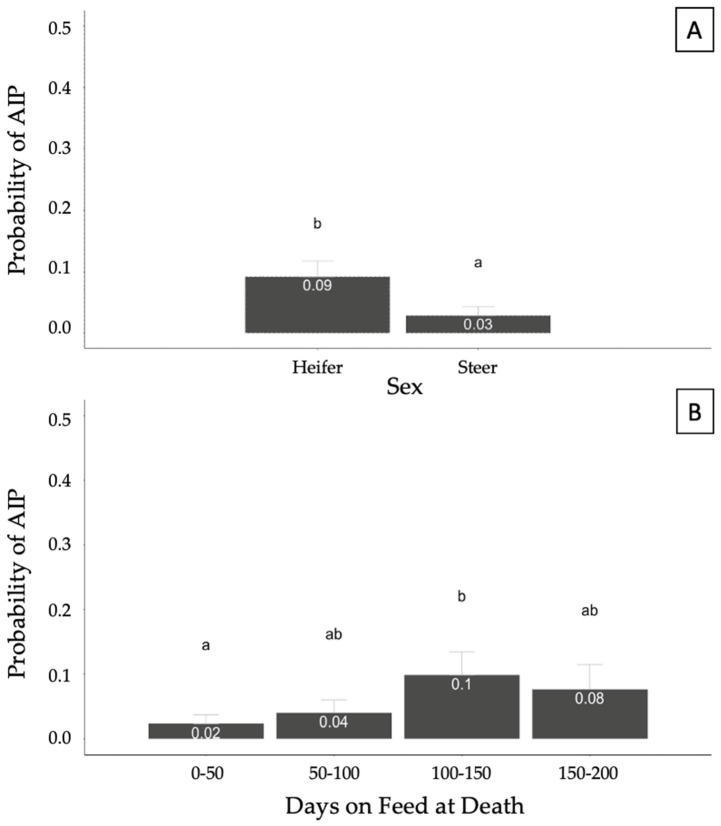
Probability of acute interstitial pneumonia (AIP) diagnosis at necropsy based on sex (**A**) and days on feed at death (**B**) in 357 cases. Results generated from a generalized mixed model accounting for the binomial response of AIP and fixed effects of sex, arrival weight, number of treatments, days on feed at death, and feedyard. Different superscript letters illustrate statistical differences in probabilities (*p* < 0.05).

**Figure 4 vetsci-10-00522-f004:**
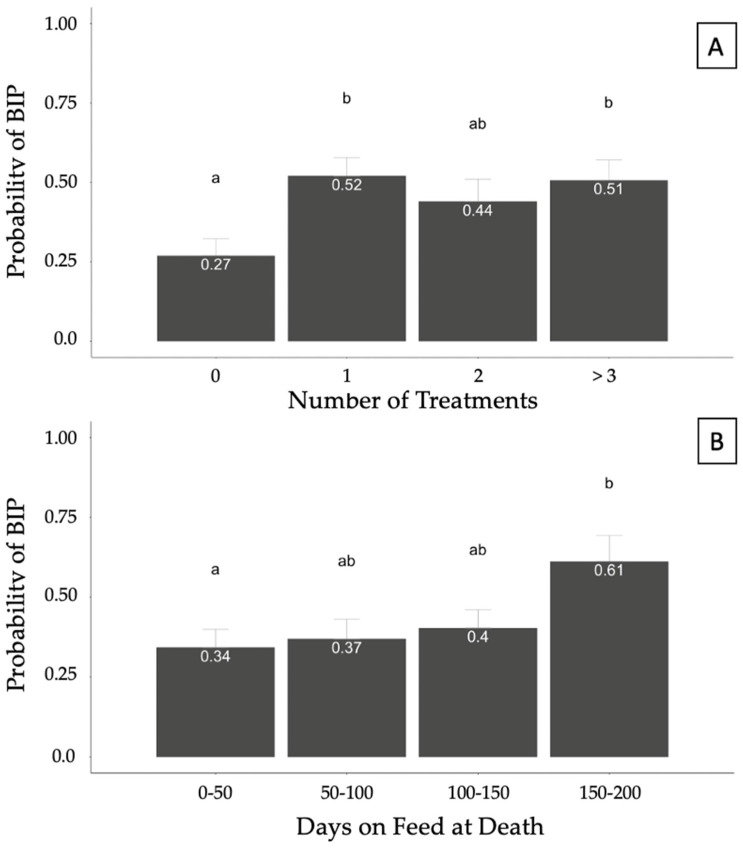
Probability of bronchopneumonia with an interstitial pneumonia (BIP) diagnosis at necropsy based on the number of treatments (**A**) and days on feed (DOF) at death (**B**) in 357 cases. Results generated from a generalized mixed model accounting for the binomial response of BIP and fixed effects of sex, arrival weight, number of treatments, days on feed at death, and feedyard. Superscript letters differing between columns illustrate statistical probability differences in pairwise comparisons (*p* < 0.05).

**Table 1 vetsci-10-00522-t001:** Descriptive count of animals according to their sex, number of treatment events, antibiotic and anti-inflammatory treatments, and arrival weight categories for each necropsy gross pulmonary diagnosis.

	Necropsy Pulmonary Gross Diagnoses			
	AIP ^1^	BIP ^2^	BP ^3^	Normal ^4^
**Sex**				
Heifer	34	98	104	16
Steer	6	44	41	14
**Treatment count, n**				
0	15	25	39	18
1	12	52	36	5
2	6	26	31	4
>3	7	39	39	3
**Antibiotic treatments, n**				
0	15	26	46	19
1	17	58	42	4
2	7	30	32	6
>3	1	28	25	1
**Anti-inflammatory treatments, n**				
0	23	68	68	19
1	11	53	39	10
2	4	15	21	0
>3	2	6	17	1
**Arrival weight categories, kg**				
<272	8	18	16	0
272–317	6	36	28	3
317–363	16	47	58	17
>363	10	41	43	10

^1^ Acute interstitial pneumonia; ^2^ bronchopneumonia with an interstitial pneumonia; ^3^ bronchopneumonia; ^4^ normal pulmonary tissue.

## Data Availability

Confidentiality and anonymity agreements with cooperating operations prohibit public disclosure of the raw data.

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
