# Peer review of "Epidemiological Factors Associated with Gross Diagnosis of Pulmonary Pathology in Feedyard Mortalities"

_vetsci, 2023, doi:10.3390/vetsci10080522_

Round 1
Reviewer 1 Report
The manuscript is well written and presented. The data from this study will be of great importance to understand the new gross diagnosis methodology proposed for respiratory disease in cattle on feed.
I just wonder if the authors were able to relate the patterns of pulmonary disease with any specific disease agent. Additionally, was the data obtained constant within the six feedyards evaluated?
Author Response
The authors would like to thank the reviewer for their comments and questions. We appreciate your time and effort to make our manuscript better. See below the reviewer's questions in black (R1) and the authors' answers in red (A).
R1: The manuscript is well written and presented. The data from this study will be of great importance to understand the new gross diagnosis methodology proposed for respiratory disease in cattle on feed.
R1: I just wonder if the authors were able to relate the patterns of pulmonary disease with any specific disease agent. Additionally, was the data obtained constant within the six feedyards evaluated?
A: A subset of the cases was sent to histopathology, and their results were reported here. All necropsy personnel were consistently trained and rotated among feedyards. All pathological findings were reviewed by a single veterinarian to help promote consistency. Cases were collected when animals were necropsied, which depended on mortality numbers and feedyard size. Feedyard effect was accounted for in the generalized logistic model.Reviewer 2 Report
The manuscrit bring a new light about pathologies observed in the bovine respiratory complex. However, I would like to suggest to include, if it was performed, a histopathological lesions suggesting a possible pathogen involved at BRD. Also association with treatments. If the animal received one or more treatment and according lesions observed which microorganism may be involved? For virus, the antibiotic treatment is innefective and consequently the animal would presented AIP, BP or BIP? Or in case of bacteria independly of the number of treatment animal received, the pathologies observed may suggest innefective of treament and consequently for One Health? And finally co-infection was it observed according to the lesions that could lead toa worsening of the disease?
Author Response
The authors would like to thank the reviewer for their comments and questions. We appreciate your time and effort to make our manuscript better. See below the reviewer's questions in black (R2) and the authors' answers in red (A).
R2: The manuscrit bring a new light about pathologies observed in the bovine respiratory complex.
However, I would like to suggest to include, if it was performed, a histopathological lesions suggesting a possible pathogen involved at BRD.
A: A subset of the samples were sent to histopathology and their results were reported here. The goal was to diagnose AIP and BP, however, pathogen involvement was not tested.
R2: Also association with treatments. If the animal received one or more treatment and according lesions observed which microorganism may be involved?
A: No cultures or etiological diagnoses were performed. Therefore, we were unable to identify with microorganisms were involved in those cases. We agreed that may have different pathogens with different presentations and it is a good area for future research.
R2: For virus, the antibiotic treatment is innefective and consequently the animal would presented AIP, BP or BIP?
A: Because we did not identify specific pathogens, it is beyond the scope of this paper to determine which etiological agents are most frequently associated with each syndrome.
R2: Or in case of bacteria independly of the number of treatment animal received, the pathologies observed may suggest innefective of treament and consequently for One Health?
A: Based on this research, we cannot identify if mortalities were the result of ineffective treatment, severe disease challenges, or infection with multiple pathogens (e.g. virus and bacteria). We agree that determining effective treatment modalities is an important part of antimicrobial stewardship and animal welfare.
R2: And finally co-infection was it observed according to the lesions that could lead toa worsening of the disease?
A: We agree co-infection could have caused worsening of the disease; however, we did not identify specific pathogens.
Reviewer 3 Report
Paper is well written. It might help to clarify the percent steer and heifer population in feedyards, since heifers seem to be over represented in mortalities.
Author Response
The authors would like to thank the reviewers for their comments and questions. We appreciate your help in improving our paper. See below the reviewer's questions in black (R3) and the authors' answers in red (A).
R3: Paper is well written. It might help to clarify the percent steer and heifer population in feedyards, since heifers seem to be over represented in mortalities.
A: A sentence was added to the results section to clarify that our enrolled population matched the population demographics of the feedyards.
“These proportions are consistent with cattle demographics within the feedyard enrolled in the study.” (line 160 and 161)